# Wearable-Based Integrated System for In-Home Monitoring and Analysis of Nocturnal Enuresis

**DOI:** 10.3390/s24113330

**Published:** 2024-05-23

**Authors:** Sangyeop Lee, Junhyung Moon, Yong Seung Lee, Seung-chul Shin, Kyoungwoo Lee

**Affiliations:** 1Department of Computer Science, College of Computing, Yonsei University, Seoul 03722, Republic of Korea; yeop.lee@yonsei.ac.kr (S.L.); kyoungwoo.lee@yonsei.ac.kr (K.L.); 2Department of Urology, Urological Science Institute, Yonsei University College of Medicine, Seoul 03722, Republic of Korea; asforthelord@yuhs.ac; 3System LSI Business, Samsung Electronics Co., Hwaseong-si 18448, Republic of Korea; sc1225.shin@dsp.yonsei.ac.kr

**Keywords:** nocturnal enuresis, in-home monitoring, wearable sensor, feature engineering, convolutional–LSTM–attention model

## Abstract

Nocturnal enuresis (NE) is involuntary bedwetting during sleep, typically appearing in young children. Despite the potential benefits of the long-term home monitoring of NE patients for research and treatment enhancement, this area remains underexplored. To address this, we propose NEcare, an in-home monitoring system that utilizes wearable devices and machine learning techniques. NEcare collects sensor data from an electrocardiogram, body impedance (BI), a three-axis accelerometer, and a three-axis gyroscope to examine bladder volume (BV), heart rate (HR), and periodic limb movements in sleep (PLMS). Additionally, it analyzes the collected NE patient data and supports NE moment estimation using heuristic rules and deep learning techniques. To demonstrate the feasibility of in-home monitoring for NE patients using our wearable system, we used our datasets from 30 in-hospital patients and 4 in-home patients. The results show that NEcare captures expected trends associated with NE occurrences, including BV increase, HR increase, and PLMS appearance. In addition, we studied the machine learning-based NE moment estimation, which could help relieve the burdens of NE patients and their families. Finally, we address the limitations and outline future research directions for the development of wearable systems for NE patients

## 1. Introduction

Nocturnal enuresis (NE), a bladder dysfunction, causes involuntary bedwetting during sleep. While the minimum frequency of bedwetting for NE patients is twice a month [1], incidents can occur more than once daily. About 20% of children of age 5 have suffered from NE [2,3]. A common NE treatment is using an alarm device that sounds upon detecting urine. Patients should attach the device to their undergarments during sleep for several weeks of the treatment. While this alarm treatment typically provides a high cure rate (e.g., 74% [4]), its failure rate is still significant. Additionally, the precise mechanisms by which the alarm treatment resolves NE remain unclear.

Clinicians agree that monitoring NE patients during the in-home alarm treatment period can contribute to enhancing NE management. It can be beneficial to continuously examine changes in NE patients from the beginning of sleep to the bedwetting moment. Naturally, the bladder volume (BV) becomes a primary examination target due to its direct relation to voiding. Traditional bladder monitoring tools used in the hospital, such as the Mindray TE7 [5], costing over USD 19,000, are often bulky and costly, making them unsuitable for continuous monitoring in daily life. In contrast, smart wearable sensors can provide viable solutions. These devices are typically lighter and smaller, making them appropriate for continuous and everyday use. Thanks to these advantages, they have been widely adopted in various areas such as continuous health monitoring, non-invasive bioelectric signal measurements, diagnostic tools, point-of-care testing, sweat and cytokine monitoring, and electrochemical detection [6]. However, to the best of our knowledge, although wearable sensor devices such as smartwatches and chestbands are commonly used to monitor various health metrics (e.g., heart rate), commercially available wearable devices for continuous bladder monitoring in daily life remain rare.

In the research field, various bladder monitoring methods have been studied based on sensor and wearable technologies using diverse types of data including ultrasound (US) [7,8,9,10,11,12], near-IR spectroscopy (NIRS) [13,14,15], and bioimpedance (BI) [16,17,18,19]. Despite these methods having expanded bladder monitoring in a portable and cost-effective manner with comparable accuracies, they appear to have limitations when applied to the long-term and continuous monitoring of pediatric patients during sleep, such as large form factors (e.g., MoUsE [8], 18.4 cm × 12.3 cm × 3.3 cm), compared to young children’s body sizes (e.g., median height of 102.5 cm for 4-year-old children [20]). In addition to the bladder monitoring, examining the heart rate (HR) and periodic limb movements in sleep (PLMS) can facilitate an in-depth analysis of NE due to their high correlations with the condition, as demonstrated in previous studies [21,22]. However, few studies have explored a comprehensive approach to monitor the bladder, HR, and PLMS of NE patients at home.

Thus, we introduced the preliminary system of in-home monitoring [23] to study the feasibility of NE patient monitoring through a wearable system. The developed system consists of three main components: wearable devices, a gateway, and a cloud server. The wearable devices, which are NEwear [24,25] and commercial wearables, collect sensor data to estimate bladder volume change (BVc), HR, and PLMS in NE patients during sleep. The gateway, a smartphone application named NEtcher, controls the wearable devices, visualizes the sensor data from the wearables, and sends the sensor data to the server named NEexplorer. From a data collection study for a few NE patients, we found that patients’ heavy movements or instances where the wearable devices were obscured by their body substantially hindered the system’s data collection capabilities. Such hindrances resulted in inaccurate data collection or data loss, thereby decreasing the reliability of subsequent data analysis (“Challenge 1”). Additionally, the preliminary system supported only monitoring capabilities without any intervention that could be beneficial for NE management. Installing an alarm to activate shortly before bedwetting incidents could significantly ease the burden for NE patients and their parents (e.g., cleanup efforts), while still retaining the effectiveness of the traditional treatment. However, there has been no approach to study the NE moment prediction using artificial intelligence (AI) techniques (“Challenge 2”). Recently, AI technology has been broadly employed in various digital healthcare applications, such as clinical deterioration prediction [26], infection detection [27], clinical decision support systems [28], energy expenditure estimation [29], and medical twins in drug delivery applications [30]. Since AI excels in the capture and analysis of nonlinear and complex patterns from high-dimensional data, it has significantly contributed to developing novel methods for diagnosis, treatment, and prevention in digital healthcare. Thus, we expect that AI would play a similar role in the context of NE patient management.

To address these challenges, we propose NEcare, an integrated system for the in-home monitoring and analysis of NE patient data, building upon the preliminary system [23]. In response to “Challenge 1”, we newly developed a data collection supervisor and NE moment detection algorithm in NEtcher. We aimed to increase the robustness of the data collection even under disturbing situations that we found with our preliminary system. To address “Challenge 2”, we conducted an NE-appropriate analysis considering the event-driven and sequential distributions of NE patient data and designed NE moment estimation algorithms using heuristic rules and deep learning methods. In addition to addressing the two challenges, we designed novel BI features to effectively estimate the BVc, offering a significant improvement over the raw BI signals used in our preliminary system.

To demonstrate the NEcare, we used the data of 34 pediatric patients (30 records collected from an in-hospital one-time study [23], and 4 collected from an in-home study for a total of 173 days). We investigated the correlations of the BVc, HR, and PLMS with regard to NE occurrence. Firstly, we examined the BVc using our proposed BI features generated from the in-hospital study data. The proposed BI features show up to a 31% higher correlation with BV (correlation = 0.67, *p*-value = 0.001) as compared to raw BI (correlation = 0.51, *p*-value = 0.017). Secondly, by analyzing the in-home study data of 78 NE days among the 173 days, we confirmed that the HR increases before the NE moment in 23 NE days. Thirdly, the clinician, who is an author of this work, confirmed that all participating patients have PLMS based on the in-home study data. Further, our designed convolutional–LSTM–attention model achieves an accuracy of up to 70.59% for one subject’s data in terms of NE moment estimation.

In conclusion, the contributions of our work are as follows: (i) introducing NEcare, an in-home monitoring and analysis system tailored for NE patients; (ii) demonstrating the system using our data collected from 34 pediatric patients (30—hospital, 4—home); (iii) investigating the correlation of our proposed BI features with bladder volume, the heart rate patterns preceding NE events, and the presence of PLMS in participants through our comprehensive data analysis; and (iv) studying the potential feasibility of AI-based NE moment estimation, which has been underexplored to date.

## 2. Existing Works on Portable Systems for Bladder Monitoring

Based on our survey, popular sensors, which have been developed or employed in the existing works for bladder monitoring, are US, NIRS, and BI sensors. Although our study focused on NE patients, the target users in this study, we comprehensively surveyed existing works not limited to only NE patients. Table 1 summarizes the existing works from our survey in terms of device type, target to be monitored, and sensor. US is a widely used sensor for monitoring urinary bladder in the hospital. Several works have adopted US sensors in a portable manner to monitor the bladder of NE patients at home [7,8,9,10,11,12]. For example, Kwinten et al. [7] developed a portable device, SENS-U (95 mm × 55 mm × 16 mm), to monitor the bladder using US data. The SENS-U was deployed for 15 NE patients at home through a single night session [7], lacking an extended-term demonstration. Fournelle et al. [8] introduced an ultrasound imaging system, MoUsE (184 mm × 123 mm × 33 mm), for automated bladder monitoring using machine learning techniques. Kuru et al. [9,10] developed a pad-type device, MyPAD (110 mm × 80 mm), based on ultrasound electronics. The form factors of these systems are significant when considering the body size of young children, as described in Section 1. Cho et al. [11] adopted a system-on-chip (SoC)-based portable ultrasound system (EdgeFlow UH10, Edgecare Inc., Seoul, Republic of Korea) to study US image-based BV estimation. They demonstrated the feasibility of AI-based BV estimation (50, 150, and 300 mL) using the US images of only tissue-mimicking phantoms. Similarly, Lee et al. [12] proposed an AI-based BV monitoring method using the US images of 250 patients from a large stationary device (SonixTouch Q+, BK Medical Inc., Peabody, MA, USA). In addition, given the high computational cost of image processing, these US image-based methods [11,12] would be limited in resource-constrained systems, such as wearable systems.

Stothers and Macnab [13] employed an NIRS device, PortaMon (Artinis BV, Leiden, The Netherlands; 83.8 × 49.2 × 17.2 mm) to measure uroflow data, including BV. Their study results were limited to the total of six voiding episodes of only two human subjects. Furthermore, Fechner et al. [14] developed an AI-based prototype method that predicted BV using both NIRS and acceleration sensor data and the elapsed time since the previous voiding. Despite the demonstrated novelty using 140 voiding cycles from 10 patients, the mean absolute error of 116.7 mL could be critical in the context of NE occurrences. In addition, Kang et al. [15] introduced an optode sensor, an NIRS-based wearable device (106 mm × 68 mm × 15 mm), to monitor BV. They demonstrated their proposed sensor using the bladder phantom, followed by a proof-of-concept test with only one human subject.

BI sensors are widely used in healthcare thanks to their capabilities of conveniently examining body components. Recently, various studies have adopted BI sensors to monitor BV [16,17,18,19,31]. Baran et al. [17] introduced a preliminary system to monitor bladder based on the images constructed using electrical impedance tomography (EIT) techniques. They used a developed belt containing 16 electrodes. Sun et al. [31] investigated BV estimation using their proposed single-layer half-circle EIT sensor. For the estimation, they conducted 3D imaging based on the data from 16 gel-type electrodes. However, the electrodes, directly attached to skin with gel-type substances, can cause inconvenience, since they might impede skin transpiration and lead to skin irritation over extended periods [32]. In addition, the image processing using BI data from many electrodes can cause high computational burdens, potentially limiting continuous monitoring in resource-constrained environments. On the other hand, there have been various bladder monitoring approaches using BI data without adopting EIT methods. Gaubert et al. [16] developed smart underwear that uses only four electrodes to monitor the bladder based on BI signals. Zhang et al. [18] proposed a belt-type device embedding seven electrodes to collect BI signals near the bladder. They investigated the machine learning-based binary classification of bladder fullness. Dheman et al. [19] developed a multi-sensor wearable system that can monitor BI, electromyography (EMG), and novel low-power charge variation (QVAR) signals. They investigated the combined use of BI with other sensor data to accurately measure BV. Compared to these works using BI signals, we developed a belt-type device, NEwear [24,25] (53 mm × 31 mm × 10 mm), which only requires two ’dry’ electrodes. We minimized the number of electrodes to be used and removed the use of gel-type substances for user convenience, at the cost of the dimensions of the data. However, since we aimed to estimate the trends of bladder volume changes (i.e., increasing or decreasing), not the exact amount of bladder volume, a higher dimension of BI data was not required.

## 3. Materials and Methods

In this paper, we present NEcare, an integrated system supporting the in-home monitoring and analysis of NE patient data, building upon our previous work [23]. As depicted in Figure 1, NEcare consists of wearable devices (NEwear [24,25] and MetaWearC [33]. Compared to our preliminary version [23], we newly developed (a) data collection supervisor and NE moment detection algorithms, (b) NE-appropriate data analysis methods considering event-driven and sequential distributions, and (c) NE moment estimation algorithms using heuristic rules and deep learning methods. While this section describes the entire system for comprehensive understanding, our emphasis is on the advancements made in this work while briefly summarizing our previous contributions. We introduce (a) in Section 3.2 and (b) and (c) in Section 3.3.

### 3.1. Wearable Devices: NEwear and Commercial Sensor Bands

To effectively monitor NE patient data at home, we utilized two wearable devices: NEwear and MetaWearC. The NEwear is a belt-type sensor device that we previously developed [24,25]. As shown in Figure 1, the NEwear collects BI and electrocardiogram (ECG) signals at 250 Hz near the NE patient’s pelvis (about 10 cm below the umbilicus). The dynamic ranges of BI and ECG sensors are from 0.1 k to 1 M Ohm and from 6 k to 12 k code, respectively. The signal-to-noise ratio (SNR) of the ECG sensor is 25 dB, and when the BI sensor is combined, the SNR is 13 dB. The reproducibility of the combined BI and ECG sensors is the r-square value of 0.99. BI signals are used to examine the BVc, and ECG signals are used to estimate the HR. The NEwear also measures resistance (R) values at 250 Hz from a moisture sensor physically attached to the patient’s underwear. In order to investigate correlations of collected sensor data with bedwetting moments and to design estimation algorithms for the moments, we first need to recognize the moments. Accordingly, we exploit R values collected from the moisture sensor to detect the urine in case of bedwetting. The moisture sensor used in this study was the Malem™ Easy-Clip© Sensor [34], which has been widely used for NE patients’ alarm treatment.

To examine PLMS, we adopted MetaWearC (MBIENTLAB, CA, USA) [33], a commercial actigraphy sensor band. As shown in Figure 1, we collect 3-axis accelerometer and 3-axis gyroscope sensor data on both patient’s ankles at 100 Hz. The NEwear and two MetaWearCs are connected to the gateway, NEtcher, using Bluetooth Low-Energy (BLE) communication.

### 3.2. Gateway: NEtcher

NEtcher, our developed Android application, manages the wearable devices to gather sensor data [23]. It delivers collected sensor data and log data to the server, NExplorer. The log data contains the information about the entire events that occur during the data collection, such as the BLE connection status and packet data length. In this work, we improved the NEtcher to support a more robust and reliable data collection by developing a data collection supervisor.

#### Data Collection Supervisor

We designed a data collection supervisor algorithm as a state machine to automate the data collection process, eliminating the need for user intervention. We determined five data collection states: Start, Normal, End, Delay, and Problem, as shown in Figure 2. The data collection supervisor conducts given tasks according to each state, while minimizing user involvement. When the user starts the data collection through the interface of the NEtcher, it begins with the Start state. At the Start state, the data collection supervisor automatically finds and connects the NEwear and the MetaWearCs. The supervisor initiates the wearable devices by sending the activation commands after it confirms that the devices are properly connected. After the supervisor receives the responses from the wearable devices (i.e., the activation is successful), the data collection state goes to the Normal state. At the Normal state, the supervisor collects the data of NE patients during sleep. During the Normal state, the data collection state is updated every 3 s for a fine-grained management. Further, the collected data is delivered to the server (i.e., NExplorer), which can avoid a total failure of the data collection due to unexpected errors. If the user attempts to end the data collection through the interface of the NEtcher, it moves to the End state. At the End state, the supervisor closes the connections with the wearable devices and uploads the collected data to the NExplorer before the termination of the data collection. If any abnormal events (e.g., a pause due to an error) occur during the Normal state, the data collection state goes to the Delay state. At the Delay state, the data collection state is updated every second for intensive examination. If the Delay state is maintained over 10 min, the data collection state goes to the Problem state. At the Problem state, the supervisor attempts to solve the problem by re-initiating the device connection or the data collection process. In case that it fails to solve the problem, the collected data are delivered to the NExplorer, and the data collection is terminated with system logs.

### 3.3. Server: NExplorer

In our prior work [23], we utilized NExplorer as a storage server for data collected from the NEtcher. In this work, we newly developed four main functions for the NExplorer to support the in-depth analysis of NE for clinicians and to check the feasibility of our prediction idea for improving conventional treatment: feature extraction, feature generation for bladder volume estimation, NE-appropriate data analysis, and NE moment estimation, as shown in Figure 3.

#### 3.3.1. Feature Extraction

Through the pre-processing stage, all the data collected from the NEtcher are interpolated and synchronized into 30 s windows. We note that the clinician recommended a 30 s window as optimal for analyzing data related to NE occurrences. Using the pre-processed data, we first extract several features related to the HR and PLMS, as shown in Table 2. We adopted features that are widely used in HR variability analyses [35] and features that are used in motion analyses through 6-axis sensors [36] for limb movement (LM). Further, we used the PLMS index, referring to Dhondt et al.’s work [37]. If the PLMS index is 5 or more per hour, clinicians determine that a patient has a PLMS disorder.

#### 3.3.2. Feature Generation for Bladder Volume Change Estimation

With regard to using the HR and LMs, few works have studied feature extraction using BI data for BV estimation. Thus, in this work, we propose novel BI features to study BV-related information, as shown in Table 2. Given that urine conducts electricity, the impedance value across the bladder is expected to decrease with urine production [38]. We confirmed such a phenomenon in our previous study with 30 pediatric patients [39]. Filling bladder with saline (i.e., BV increase) causes a decrease in the BI value collected near a pediatric patient’s pelvis. However, we also discovered that baseline BI values are different for each individual. In addition, body posture changes baseline BI values even for the same patient. Thus, we needed to examine BI values in various dimensions instead of using raw values only. Accordingly, in this work, we introduce novel BI features focusing on the difference between before and after a certain period of time has passed. Firstly, we designed a BI feature named BI delta (BID), as shown in Equation (Equation 1). Since heavy body movements normally cause a huge difference in the BI values in our preliminary study, we empirically set an upper bound (D_upper_ to remove such differences. The accumulated BI difference values for a certain period of time (0–T) can represent the BV change in that duration. In addition, we designed BID_decrease_ to focus more on the decreasing trends of BI values. As shown in Equation (Equation 2), it accumulates the non-negative difference (nnD) of BI values. In addition, we designed the delta of BID (DBID) and clutter (Clt) to consider the speed of the BVc and background information of the BI signals, respectively. Note that clutter typically represents background information especially in radar signal processing [40].
$$D_{i}=\left\{\begin{array}{cc} BI_{i}-BI_{i-1}, & \mathrm{if}\;|BI_{i}-BI_{i-1}|\leq D_{\mathrm{upper}}\\ 0, & \;\mathrm{otherwise} \end{array}\right.$$
(1)$$BID_{i}\left(T\right)=\sum_{i=1}^{T}D_{i},$$
$$nnD_{i}=\left\{\begin{array}{cc} BI_{i}-BI_{i-1}, & \mathrm{if}\;0\leq BI_{i-1}-BI_{i}\leq D_{\mathrm{upper}}\\ 0, & \;\mathrm{otherwise} \end{array}\right.$$
(2)$$BID_{\mathrm{decrease}i}\left(T\right)=\sum_{i=1}^{T}nnD_{i},$$

#### 3.3.3. NE-Appropriate Data Analysis

In this study, we approached our NE patient data analysis from two perspectives: event-driven and sequential. From one perspective, NE patient data should be analyzed based on the appearance of bedwetting events (event-driven). For example, Pretlow [1] and Bader et al. [21] studied changes in the BV or HR near NE moments. As depicted in the upper part of Figure 4, an event-driven analysis can compare how features show different values at the start, middle, and end of the entire data collection period. From the other perspective, NE patient data should be investigated according to the timeline in a sequential manner. For example, Dhondt et al [22,37] studied the relationship between the sleep sequences and NE, using the PLMS index and sleep stages across the entire sleep duration. As shown in the lower part of Figure 4, features are sequentially examined from the start to the end of the sleep duration.

#### 3.3.4. NE Moment Estimation

Conventional alarm treatments alert NE patients after they committed involuntary bedwetting using sound alarms. If we can awaken patients before bedwetting occurs, it could reduce the cleaning burden on the patient’s family while preserving the effectiveness guaranteed by conventional treatments. Consequently, in this work, we introduced five NE moment estimators to predict when NE might occur. We designed four rule-based estimators, RE_B_, RE_BL_, RE_BH_, and RE_ALL_, and a deep-learning-based estimator (DE). As shown in Table 3, the rule-based estimators require different combinations of input data (B: BV, H: HR, L: LM). Since BV is an important component of the human voiding system, all estimators take BV as an input source. In this work, BV changes are estimated by our designed BI features. Accordingly, if the BI features satisfy certain thresholds, all rule-based estimators assume that the bladder is almost full. In addition, we studied the effectiveness of the HR and LM features for NE moment estimation using different combinations of input sources (RE_BL_, RE_BH_, and RE_ALL_). The estimators examine whether the HR increases or LM appears near NE moments. Eventually, all rule-based estimators estimate possible NE moments according to whether given features satisfy certain conditions. We determined the size of the data window for the entire estimators to be 10 min and to slide each window by 30 s.

To design the DE, labeled data were essential for supervised learning. However, since NE occurs once or twice during the entire sleep period (about 8 h), it is difficult to gain balanced data. Further, it is difficult to categorize the entire sleep data into discrete classes in terms of NE occurrence. Therefore, we proposed using linear regression models to represent the probability of NE occurrence as time passes, as shown in Figure 5. For NE days, the probability of NE is 100% at the point where NE has indeed occurred, and it gradually decreases as the point moves far away from the NE moment (heading to 0). For dry days (i.e., days when NE did not occur), we determined that the highest probability of NE occurrence appears at the end point of the data collection, and it is 70% based on a clinician’s advice. We normalize the NE probability into the range from 0 (0%) to 1 (100%). We determined that if the normalized NE probability increases over 0.85, NE will occur soon.

With our designed NE probability models, we developed the DE using a convolutional neural network (CNN), long short-term memory (LSTM), and an attention mechanism, as depicted in Figure 6. Firstly, in order to effectively process our input data from different sources, the BV, HR, and LMs, we adopted the a multi-dimensional CNN structure. CNNs are effective for image feature extraction, capturing spatial hierarchies where higher-level features are composed of lower-level ones. They are widely used for image and video recognition, natural language processing, and other applications where spatial information is critical (e.g., motion detection areas [41]). Wang et al. [42] proposed the multi-channel CNN structure for multi-channel EEG signals originating from a single source (the brain). The proposed CNN structure has been demonstrated to be advantageous for analyzing EEG signals, especially when kernels encompass the entire input dimension. Adopting the same idea, we designed a multi-dimensional structure consisting of three separate CNNs for HR features, BV features, and movement features, as shown in Figure 6. Secondly, we added a bi-directional LSTM layer to learn the sequential aspects of the integrated feature vectors generated from the CNN layer. The bi-directional LSTM model processes data in two opposite directions with two separate hidden layers, which are then fed forward to the same output layer [43]. This allows the entire network to learn both backward and forward information about the sequence data. Finally, we use a attention mechanism to enable the models to focus on different parts of the input sequence when predicting an output, essentially deciding at each step where to look in the input sequence for the most relevant information [44,45]. This can significantly help the prediction network learn which parts among the entire sequence of data are more important than others in terms of the prediction.

## 4. Experiments

We demonstrated our developed system, NEcare, using patient data from both in-hospital and in-home studies. In this section, we introduce the details of both studies.

### 4.1. In-Hospital Data Collection in Urodynamic Study

In our previous work [39], we conducted data collection using the NEwear and the NEtcher during a urodynamic study (UDS) [46] to check the feasibility of pediatric patient monitoring and to analyze correlations between the BV and BI. The UDS is a clinical test in which medical staff insert a catheter into a patient’s urethra, filling the bladder with saline until it is full. The study received approval from the Institutional Review Board (IRB) of Severance Hospital (4-2018-0500), and all participants gave their consent. Throughout the study with 30 pediatric patients, we discovered that the BI collected near the pelvis decreased as the bladder increased because of the injection of saline. In this work, we used the patient data collected from the UDS to demonstrate our designed BI features since we can compare the proposed BI features and the amount of increased bladder volume.

### 4.2. In-home Data Collection of NE Patients

We deployed our NEcare to NE patients who were recommended to be involved in the conventional alarm treatment according to a clinician’s diagnosis. Note that our previous works did not explore the feasibility of NE moment estimation, and to our knowledge, no other studies have delved into this concept. Accordingly, we decided that we needed to study the feasibility in this work and then deployed the system to support real-time NE moment estimation in future works. For the data collection, the NE patients wore our wearable devices as instructed, with their parents overseeing the entire process for approximately 8 weeks. This study was approved by the IRB of Severance Hospital (4-2017-1214). We conducted the data collection, and four pediatric NE patients (P1–P4) gave their consent to finish the collection.

## 5. Results

As detailed in Table 4, we successfully collected data from four NE patients during sleep over 173 nights. We describe our experimental results in three different subsections. First, we demonstrate the correlation of our proposed BI features with the BV changes. Second, we present the analysis results with our experimental data to investigate the relationship between NE and examined data. Third, we present the estimation of the NE-possible points using the five NE estimators that we designed.

### 5.1. Correlations between BV Changes and the Proposed BI Features

To demonstrate our proposed BI features in terms of correlations with the BVc, we used the pediatric patients’ data collected in our in-hospital study (Section 4.1). The data from the in-hospital study are suitable for validating our BI features, as they encompass both BI values and the actual volume of the saline injected into the bladder. Since all the patients voided before the data collection began, we considered the maximum amount of injected saline to be equal to the volume of each patient’s bladder capacity. Using the data, we examined the correlations of the raw BI values and our designed BI features with the BV (from 0 cc to the subject’s bladder capacity). Among these, the BID exhibits the highest Pearson Correlation Coefficient (PCC) at 0.67 (*p*-value: 0.001). It is 31% higher than the PCC value of the raw BI (PCC: 0.51, *p*-value: 0.017). Figure 7 shows the distributions of the BI and BID values (x-axis) according to the bladder capacity values (y-axis). Given that the BID correlates most strongly with the measured BV, we used the BID to represent BV changes when analyzing the in-home study data. Note that we did not measure the BV in our in-home study. This is because such a measurement is not mandated in conventional alarm treatments and could impose significant burdens on patients. Figure 8 shows the trends of the BI, BID, and BID_decrease_ values collected during one NE day of a subject. Over time, our proposed BID and BID_decrease_ exhibit a gradual decreasing trend. However, the raw BI values change very sharply at some points (mostly because of heavy movements), and it is difficult to understand the trend of these values.

### 5.2. Investigation of BV, HR, and PLMS Regarding NE

Using the in-home study data from four NE patients (Section 4.2), we explored expected trends related to NE occurrence as identified in the literature: BV increase, HR increase, and PLMS appearance.

#### 5.2.1. BV Increase

Figure 9 shows the BID distributions based on our event-driven and sequential-driven analyses. Figure 9a,c,e,g show the BID distributions with the start, middle, and end points of the data collection of each subject. As shown in Figure 9a,c,e,g, BID values at the end points are always lower than those at the start or middle points for both NE and dry days. This confirms that BV increases during sleep according to our demonstration on the relationship between BID decrease and BV increase (Section 5.1). Further, in Figure 9b, the overall slope of the BID distributions over time on NE days is relatively sharper than the one on dry days. This might be related to what [1] reported in that the speed of bladder filling is fast on NE days. However, it is difficult to find similar patterns in Figure 9b,f,h.

#### 5.2.2. HR Increase

We discovered that among a total of 78 NE days, the HR increases before the NE moments in 23 days (29%). Given that the HR can rise during sleep due to factors other than just NE occurrence, we analyzed HR fluctuations throughout the entire sleep duration. Figure 10 shows HR distributions according to our event-driven and sequential analyses. Figure 10a,c,e,g show the HR distributions with the start, middle, and end points of the data collection of each subject. As shown in Figure 10a,c, P1’s and P2’s HR at the end points of NE days are lower than those of dry days (i.e., non-NE days). However, as shown in Figure 10e,g, P3’s and P4’s HR at the end points of NE days are higher than those of dry days. Figure 10b,d,f,h show the HR distributions for sequential analyses for each subject. In Figure 10h, P4’s HR of NE days mostly stays in a range of higher values than that on dry days. However, the other subjects’ HR shows different trends compared to P4’s HR.

#### 5.2.3. PLMS Appearance

The clinician confirmed that all participants have a PLMS disorder. Table 5 shows the average PLMS index for each subject on a dry day, an NE day, and the total day (dry day + NE day), separately. If the PLMS index is 5 or more, we can conclude that an NE patient has a PLMS disorder. The average of the PLMS index of our subjects was 15.59, which is similar to that of the NE patients that Dhondt et al. [22] reported on (13.9).

### 5.3. NE Moment Estimation Results

We used our five designed estimators to predict potential NE moments. Figure 11 shows estimation results using P3’s data. Black-colored parts in each bar graph show the estimated moments. As shown in Figure 11a, RE_B_ mostly estimates NE-possible moments mostly after a certain point on the same day because RE_B_ generates the estimation result according to a single threshold. As shown in Figure 11b, RE_BL_ generates similar results to RE_B_. Since young patients usually move more than adults, it seems that considering whether limb movements appear or not does not make a huge difference between the estimation results of RE_B_ and RE_BL_. As compared to RE_B_ and RE_BL_, RE_BH_, RE_ALL_, and the DE generate significantly fewer NE-possible moments, as shown in Figure 11c–e. Interestingly, Figure 11e shows that the DE generates NE-possible moments only on NE days.

Figure 12 shows the overall performances of the five NE moment estimators we designed using the entire data of four NE patients. The four rule-based estimators (RE_B_, RE_BL_, RE_BH_, and RE_ALL_) determine NE moments based on specific rules. In contrast, the DE first estimates the possibility of NE and then determines the NE moment based on this estimation. Thus, investigating estimation results at the data window level are not compared fairly. We compared the estimation results of the five estimators at the day level since we have the actual ground truth about if a day was an NE day or not. The DE shows the highest accuracy (71%) for P1’s data. Excluding P4, the DE shows higher accuracy and recall values than the other estimators.

### 5.4. Discussion

This work presents NEcare, a wearable-based system that supports the in-home monitoring and data analysis of NE patient data. Using our developed data collection supervisor, NEcare successfully collected data over 173 nights from four pediatric patients (in-home data). Leveraging the in-home data with the in-hospital data (30 patients), we demonstrated that our proposed BI features are effectively used to show the changing trends of the BV. Further, we observed the anticipated HR increase and PLMS appearance from our participating patients at home through NEcare, which were reported in previous works mostly conducting in-hospital or in-laboratory examinations [21,22]. In addition, our work also investigated the AI-based NE moment estimation approach with real-world patient data. Figure 13 shows a summary of the strengths, weaknesses, opportunities, and threats of our NEcare system. The NEcare system increases user convenience by combining BI and ECG sensors into a single device (i.e., NEwear) and avoiding gel-type electrodes. Additionally, the size of NEwear seems acceptable even for kids. Moreover, since NEcare supports BI, ECG, and accelerometer sensors, we can monitor multiple symptoms related to NE, such as increases in the BV and HR and the appearance of PLMS. NEcare also has a novelty since it contributes to the construction of the foundation for AI-based NE moment estimation. However, since NEwear utilizes only two electrodes, it is insufficient for estimating the exact BV. In addition, the least number of electrodes may lead to inaccurate data collection under some situations, since the dimensions of BI signals are low. Finally, the low performance of NE moment prediction requires further improvements for the practical application of NEcare to the real world. Despite these weaknesses, we believe that there are opportunities for our NEcare system, including in-home and long-term bladder monitoring, as a contribution to novel treatment methods for bladder dysfunctions, and its utilization for novel studies related to data from the abdomen. Of course, there are potential threats to the NEcare system, which limit or hinder its broader usage in various fields. These include smart underwear with comprehensive sensors, the development of novel non-contact sensors, and the appearance of advanced AI models that estimate the BV using only widely used sensors (e.g., PPG).

To achieve the practical use of our NEcare, several limitations must be addressed in future works. Firstly, the number of NE patients in the in-home study seems small, which might lead to a lack in sufficiently representing the entire NE patient population. We note that the pandemic made recruiting pediatric patients challenging, as parents are hesitant to bring their children to the hospital. However, our data collection from patients’ home environments for a total of 173 days cannot be underestimated. To the best of our knowledge, our work is the first approach to design an in-home monitoring system that can collect not only bladder-related data but also heart rate and body movement data. Accordingly, even though our experimental results in this work are insufficient to conclusively demonstrate the system, we believe that the results can contribute to various future works related to monitoring and intervention for NE patients. Based on what we learned from this study, we will improve our system, NEcare, and then we will deploy the improved system to a broader population of NE patients.

Secondly, the limited dataset can also raise concerns on the establishment of the benefits from our proposed system, NEcare. NEcare has some benefits in NE patient monitoring, such as a smaller form factor compared to that of previous systems [7,8,9,10,15], the use of less electrodes compared to those in works using BI sensors [16,17], and avoiding high-computational image processing, which was required in Dunne et al,’s work [17,31]. Despite the potential of NEcare in NE patient monitoring at home in a more convenient and effective manner, we still need to validate the benefits of NEcare on a larger number of NE patients under various environments in future works.

Thirdly, despite NEwear being smaller than the sensor devices used in existing works in this field [7,8,15], it still needs to be improved to being easily acceptable for pediatric children. We observed that all four NE patients from our in-home data collection sometimes expressed inconvenience with wearing NEwear. The NEwear, positioned near the abdomen, may be less user friendly than other commercial wearables like wristbands (e.g., vívofit^®^ jr. 3—Garmin [47], a kid’s fitness tracker (ages 4+)). It is uncommon for individuals to wear devices attached to the skin near the waist, making it challenging to ensure a snug fit for the NEwear. However, a loose fit of NEwear could compromise the data accuracy, potentially degrading the performance of the monitoring and intervention for NE patients. To resolve this issue, various approaches should be investigated, such as designing mechanisms to process noisy signals from a loose-fitting NEwear and adjusting the form and the texture of the NEwear. Given that smartwatches have been utilized for various health-related purposes such as step counting, heart rate monitoring, and activity-level estimation, we believe that novel belt-type wearables on the waist may open new paths to delivering various types of health services.

Fourthly, to enhance the analysis and estimation capabilities of NEcare, we may need to incorporate additional sensors or tools from different modalities. While we identify several anticipated trends related to NE occurrences, such as increases in the HR and BV, these patterns are evident in only some participants and were not consistent throughout the entire study duration. These inconsistencies might be attributed to individual variations and daily fluctuations. In these situations, effectively utilizing multiple types of data collected from different sources may complement the limited observation of the patients. For example, radar-based monitoring methods can be adopted for this purpose, which have been studied to estimate vital signals such as the heart rate and breathing rate without the need to wear devices attached to the skin [48].

Finally, we need to address the limited performance of NE moment predictions in future works. As highlighted in Section 3.3.4, there is a pressing need to address the data imbalance between NE moments and non-NE periods. Although our regression model offers a solution by representing NE possibility, it is constrained by the absence of a ground truth for validating the generated probabilities. Implementing data augmentation techniques tailored for NE could mitigate this imbalance, ultimately enhancing the accuracy of NE moment estimation. Although there are several predictive approaches for NE patients [9,10,49,50], we believe that our work has a novelty compared to these works. Franco and Coble [50] proposed a novel bedwetting alarm system utilizing real-time heart rate variability (HRV) analysis and machine learning in order to wake users before wetting occurs. However, as we discussed above, a prediction based on a single data source may totally fail in cases where the data source is not available because of several issues including device failure and the loosened wearing of the device. Jönsson et al. [49] utilized a mobile application to gather nightly patient data, which they analyzed using a random forest machine learning model to predict early outcomes of enuresis alarm treatment. Although utilizing the daily data from the user input (i.e., active data), not the sensor data (i.e., passive data), may suggest another source of data in this field, their goal is not consistent with ours, which is predicting the moments of bedwetting. Moreover, since requiring users to log the data manually is a significant burden, we need to carefully consider this approach in future works. Although Kuru et al. [9,10] investigated AI-based NE alarm generation by estimating the fullness of the bladder using US sensor data, the device’s form factor is significant, and they lacked a demonstration with human subjects. Overall, compared to these existing approaches for bedwetting prediction, our NEcare would contribute to not only the NE moment prediction aspect but also in revealing the unknown dimension of NE itself by supporting ECG, BI from the bladder, and body movement data from the ankles altogether.

## 6. Conclusions

In this study, we introduced NEcare, an integrated system designed for the in-home monitoring and analysis of NE patient data. While the preliminary version of this system was established in our prior research, we have now enhanced it by incorporating various new functionalities. This refined system not only facilitates a deeper analysis of NE patient data but also explores the potential of a novel approach, that is, NE moment estimation, to complement conventional treatment procedures. Our system’s efficacy was demonstrated using data from pediatric patients, both from in-hospital studies involving 30 patients with bladder disorders and in-home studies with 4 NE patients. We successfully validated expected trends in the BV, HR, and PLMS in real-world settings, informed by the existing literature. Notably, our proposed BI features exhibit a strong correlation with bladder volume changes. Our exploration into NE moment estimation, employing both rule-based and deep learning-based algorithms, yielded an accuracy of up to 71% for one patient’s data on a daily basis. This work focused on examining the feasibility of the proposed wearable system in NE patient monitoring at home. To conclusively demonstrate our system, future works are required for a broader population of NE patients.

## Figures and Tables

**Figure 1 sensors-24-03330-f001:**
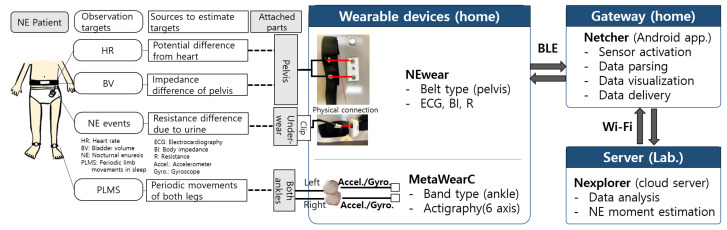
Overview of our monitoring and analysis system, NEcare, that consists of wearable devices, a gateway, and a server to support the in-home treatment of NE patients.

**Figure 2 sensors-24-03330-f002:**
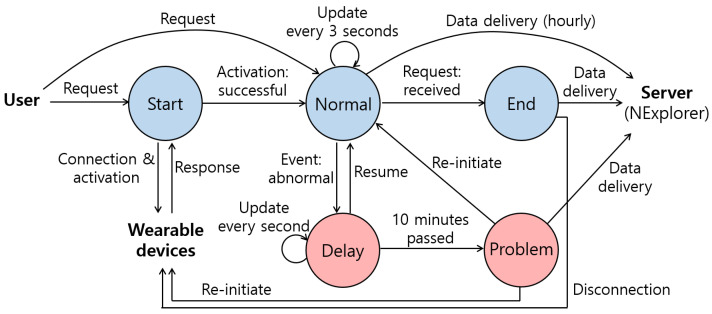
Overview of the data collection supervisor in the NEtcher, which supports a more robust and reliable data collection using our wearable devices (info.: information, addr: address).

**Figure 3 sensors-24-03330-f003:**
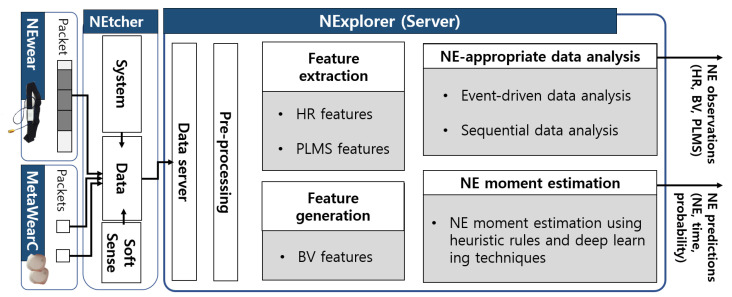
Overview of the NExplorer, including the feature extraction, feature generation, NE-appropriate data analysis, and NE moment estimation functions.

**Figure 4 sensors-24-03330-f004:**
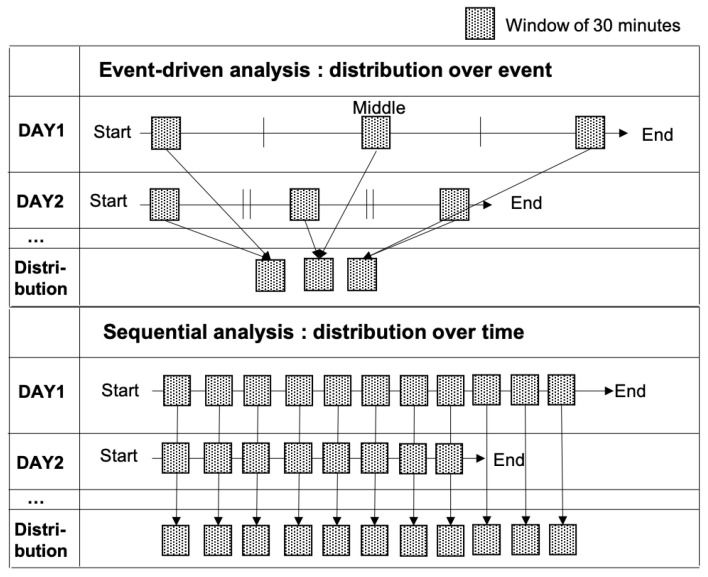
Distributions of 30-second data windows for event-driven analysis (**upper**) and sequential analysis (**lower**).

**Figure 5 sensors-24-03330-f005:**
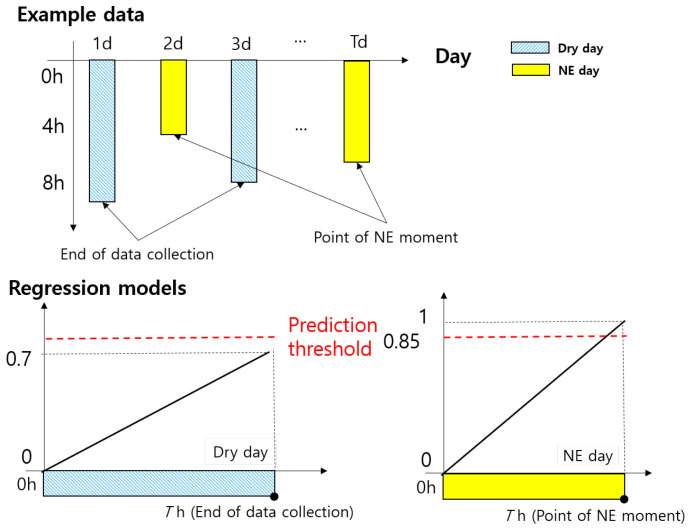
Example data of NE and dry days and the proposed linear models used to represent NE possibility according to the dry day or the NE day.

**Figure 6 sensors-24-03330-f006:**
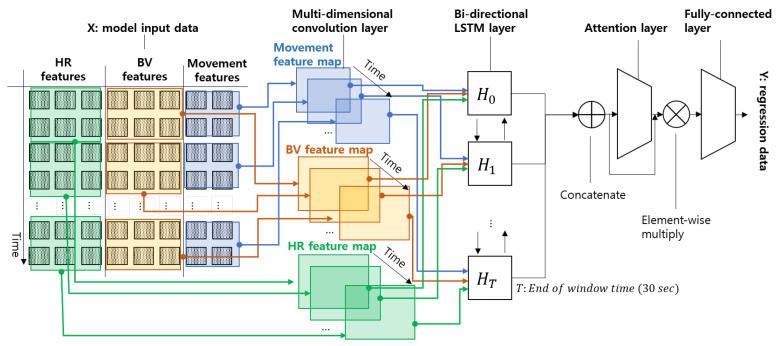
The architecture of the proposed attention-based neural network, which can individually interpret input data through multi-dimensional convolution layers and improve the effectiveness of the estimation accuracy.

**Figure 7 sensors-24-03330-f007:**
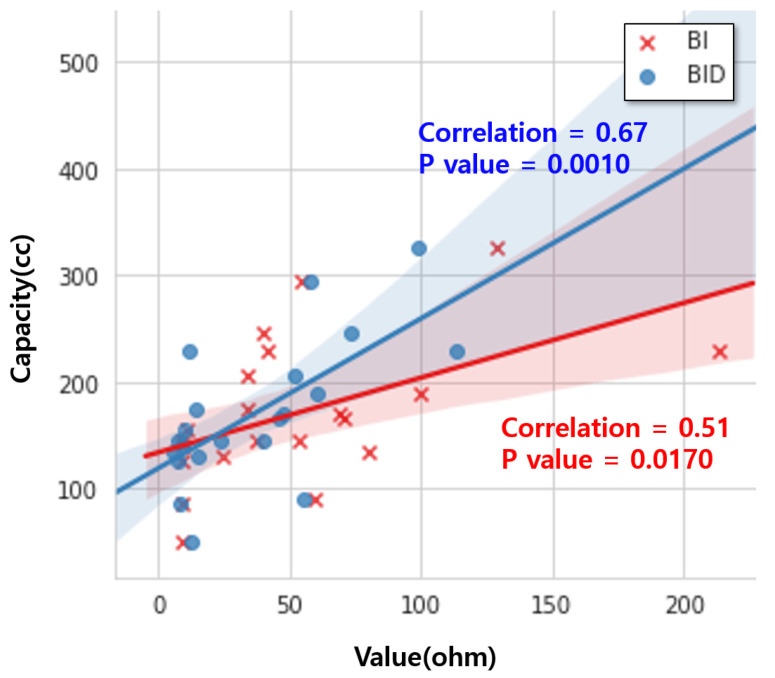
Distribution and correlation of BID and BI values versus BV capacity.

**Figure 8 sensors-24-03330-f008:**
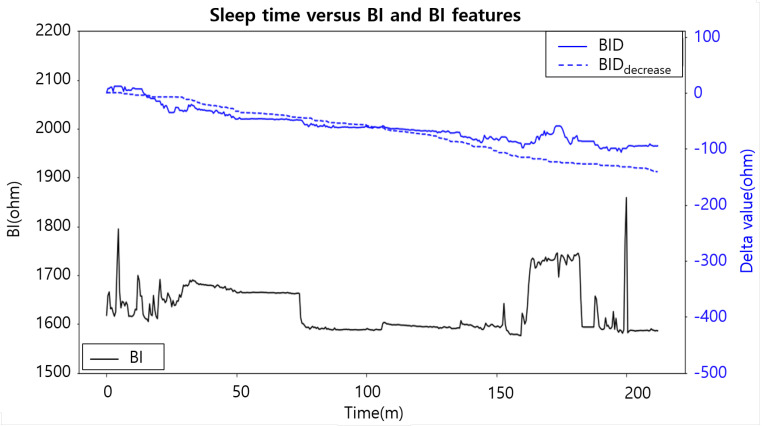
Trends of the BI, BID, and BID_decrease_ values of one patient’s NE day. The left y-axis represents the values of the BI, while the right y-axis shows the values of the BID and BID_decrease_.

**Figure 9 sensors-24-03330-f009:**
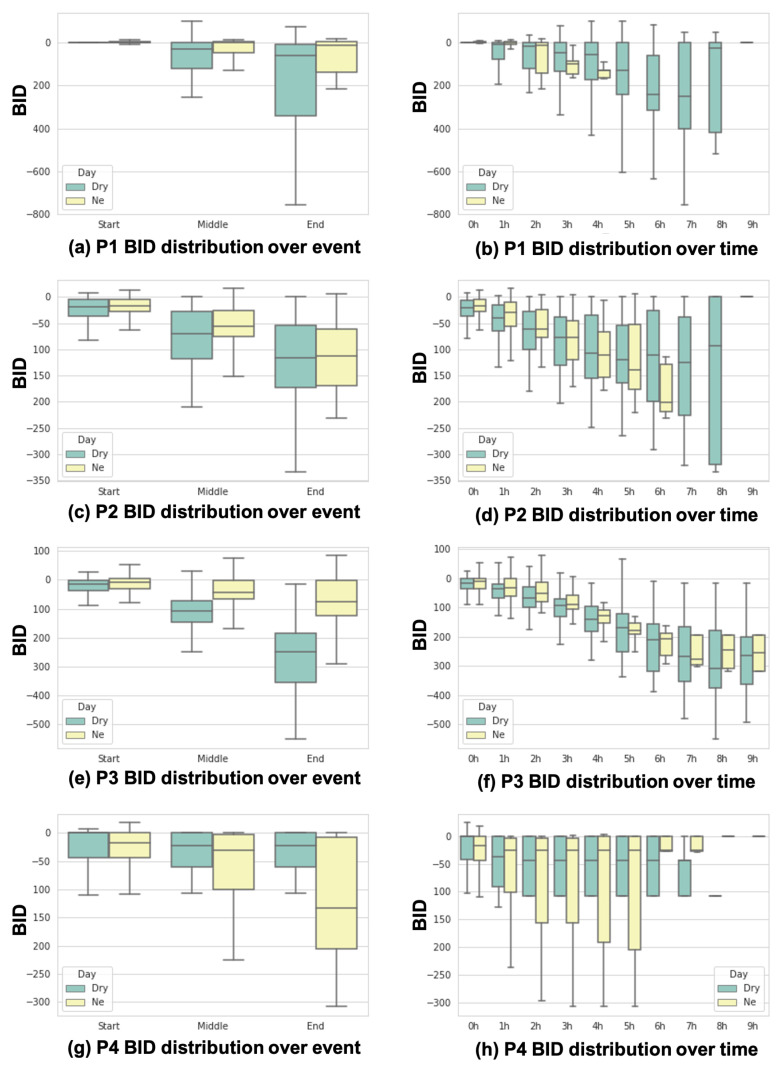
Event-driven and sequential analysis results of BID for all subjects.

**Figure 10 sensors-24-03330-f010:**
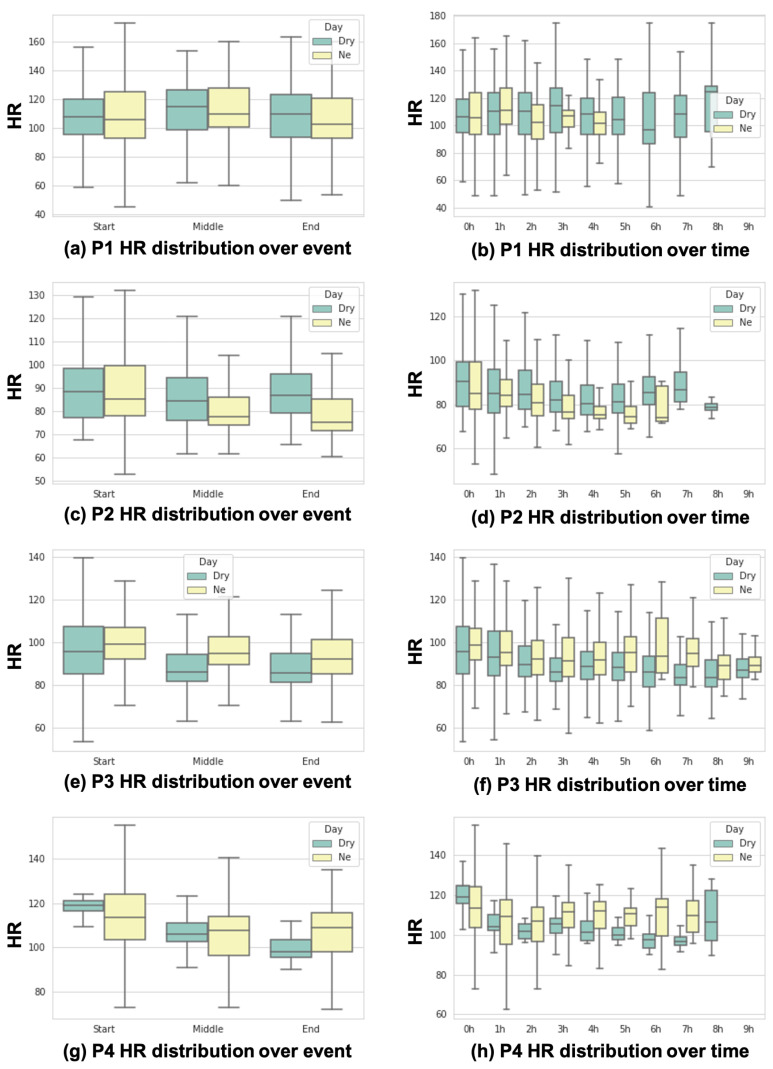
Event-driven and sequential analysis results of HR for all subjects.

**Figure 11 sensors-24-03330-f011:**
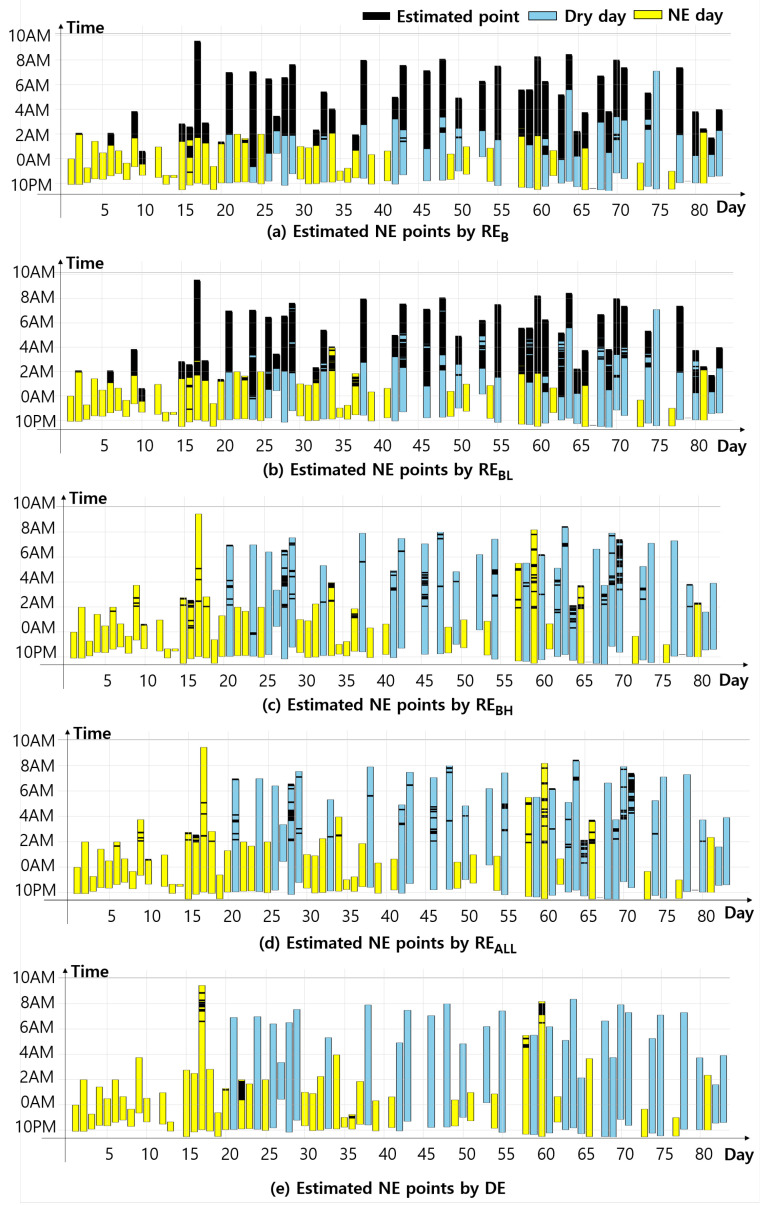
Estimation points that our designed estimators generated for P3.

**Figure 12 sensors-24-03330-f012:**
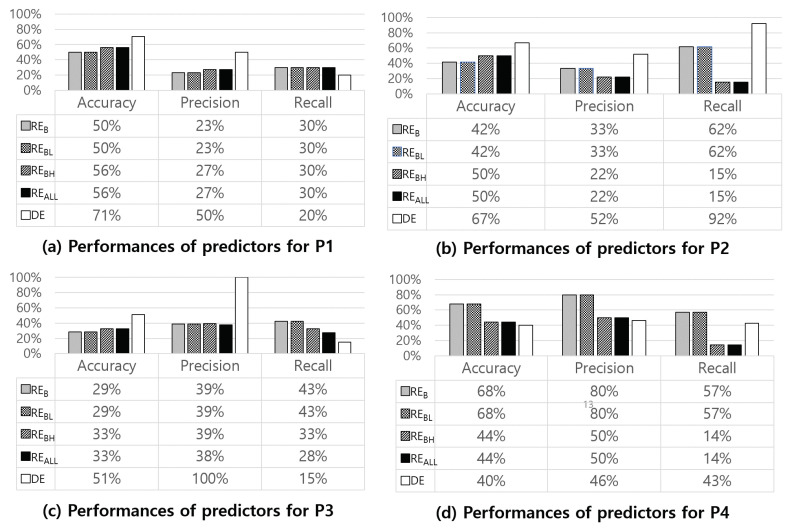
Comparison of NE moment estimation performance in terms of accuracy, precision, and recall for the five estimators we designed.

**Figure 13 sensors-24-03330-f013:**
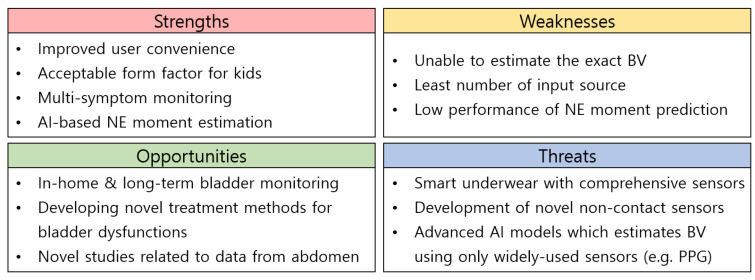
SWOT analysis of the proposed NEcare system.

**Table 1 sensors-24-03330-t001:** Comparison of existing works on portable devices and systems that can be used for NE patient monitoring (US: Ultrasound, NIRS: Near-IR spectroscopy, BI: Bioimpedence, EMG: electromyography, QVAR: novel low-power charge variation).

Portable Systems	Device type	Target	Sensor
SENS-U [7]	Pad	BV	US
Gaubert et al. [16]	Smart underwear	BV	BI
MoUsE [8]	Pad	BV	US
Baran et al. [17]	Belt	BV	BI
PortaMon (used in [13])	Placed by fingers	Uroflow data (e.g., BV)	NIRS
Fechner et al. [14]	Attachable	BV	NIRS, Acc
MyPAD [9,10]	Pad	BV (Phantom)	US
EdgeFlow UH10 (used in [15])	Scanner	BV (Phantom)	US
Optode sensor [15]	Portable	BV	NIRS
Sun et al. [31]	Attachable	BV	BI
Zhang et al. [18]	Belt	BV	BI
Dheman et al. [19]	Patch	BV	BI, EMG, QVAR
SonixTouch Q+ (used in [12])	Stationary system	BV	US
Preliminary of NEcare [23]	Belt	HR, BV	ECG, BI
NEcare (this work)	Belt, ankle bands	HR, BV, PLMS	ECG, BI, Acc, Gyro

**Table 2 sensors-24-03330-t002:** Extracted and generated features used to examine BV, HR, and PLMS that are related to NE occurrence.

Category	Features
BV (5)	BI, BID_decrease, BID, DBID, Clt
HR(V) (9)	HR (max, mean, min, std), nni 20, nni 50, pnni 20, pnni 50, range nn
LMs (7)	Trend of angle differences (Left, Right), movement magnitude (Left, Right), motion counter (Left, Right), PLMS index

**Table 3 sensors-24-03330-t003:** Rule-based estimators (REs) and a deep learning-based estimator (DE) for estimating NE moments with various combinations of input data sources.

Estimators	RE_B_	RE_BL_	RE_BH_	RE_ALL_	DE
BV	O	O	O	O	O
HR	X	X	O	O	O
LM	X	O	X	O	O

**Table 4 sensors-24-03330-t004:** Summary of each subject’s participation in our study.

Subjects	The Entire Treatment	Data Collection	NE Days
P1	39 days	34 days	10 days
P2	46 days	38 days	13 days
P3	84 days	73 days	41 days
P4	47 days	28 days	14 days

**Table 5 sensors-24-03330-t005:** Average of PLMS index for each subject according to dry days, NE days, and the total days of the data collection.

Subject	Dry Day	NE Day	Total Day
P1	13.06	19.88	17.87
P2	9.07	15.45	13.41
P3	10.23	6.03	8.45
P4	21.17	25.97	22.61

## Data Availability

Unfortunately, the dissemination of the raw data underpinning our study is not feasible. In the course of conducting the research presented in this manuscript, we collected datasets that encompass highly sensitive information, collected both from patients’ residences and the hospital. Consequently, sharing these datasets in any form is significantly challenging. We appreciate the academic community’s understanding of these constraints and are committed to upholding the highest standards of data privacy and security.

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
