# Peer review of "Wearable-Based Integrated System for In-Home Monitoring and Analysis of Nocturnal Enuresis"

_sensors, 2024, doi:10.3390/s24113330_

Round 1

Reviewer 1 Report

Comments and Suggestions for Authors

This paper states that " NEcare consists of wearable devices including our developed NEwear, a gateway named NEtcher, and a server named NExplorer. NEcare properly collects electrocardiograph, body impedance (BI), 3-axis accelerometer, and 3-axis gyroscope sensor data that are used to examine bladder volume (BV), heart rate (HR), and periodic limb movements (PLMS) in sleep appearance." and “we study NE moment estimation by using heuristic rules and deep learning techniques, which could help relieve burdens of NE patients and their families.” which seems to be a very interesting result. In particular, it is interesting that the exploration into NE moment estimation, employing both rule-based and deep learning-based algorithms, yields an accuracy of up to 71% for one patient’s data on a daily basis. However, before acceptance of the paper, the authors need to add and improve some missing information.

 First of all, in this study, you verified the usefulness of the developed device by comparing data in 4 NE patients, but we believe that 4 patients are not enough to calculate a statistical difference. In the Discussion section, you say the following: “While we identify several anticipated trends related to NE occurrences, such as increases in HR and BV, these patterns are evident in only some participants and not consistently throughout the entire study duration. These inconsistencies might be attributed to individual variations and daily fluctuations.”

As you mentioned here, the scientific data may be influenced by individual reasons such as individual differences based on data from a small number of 4 subjects, so the conclusions of your paper are not sufficiently proven. I don't think this is possible, but it is important to explain this point carefully. Please add your information.

 Your research is very interesting, but I think it is necessary to clarify the explanation regarding data reliability. Please clarify this point.

Author Response

We sincerely thank you for acknowledging the strengths of our manuscript and for suggesting an intriguing area for enhancement. In response to your valuable suggestion, we clarified what we have found in this work (i.e., checking the feasibility of observing NE-related trends from our four patients) and what we need to do in future works to generalize and conclusively demonstrate the outcomes of this work. For this, we revised the last paragraph of Introduction (p3), the second paragraph of Discussion (p15-16) and Conclusion (p18-19).

Reviewer 2 Report

Comments and Suggestions for Authors

Summary:

The Authors present a monitoring and analysis system for nocturnal enuresis (NE) with an aim to come up with the algorithm that would raise alarm before NE occurs.  The proposed system is evaluated on 34 patients using BV, HR and LM as the input data for NE moment estimation, based on total of four estimators.

Comments:

Overall, I find that the content of the paper is well structured and suitable for journal publication as it conveys scientifically important insights to the field of NE wearable devices.

 Following  corrections are advised in this round of review, in order to improve the completeness of the paper.

1.

Introduction misses some recent related work, and the comparison of the proposed system to the existing related systems, such as

a)      Kuru, Kaya, et al. "Smart Wearable Device for Nocturnal Enuresis." 2023 IEEE EMBS Special Topic Conference on Data Science and Engineering in Healthcare, Medicine and Biology. IEEE, 2023.

b)      Franco, Israel, and Jon Coble. "Initial outcomes using a novel bedwetting alarm (Gogoband®) that utilizes real time artificial intelligence to wake users prior to wetting." Journal of Pediatric Urology 19.5 (2023): 557-e1.

c)      Jönsson, Karl-Axel, et al. "Improving the efficacy of enuresis alarm treatment through early prediction of treatment outcome: a machine learning approach." Frontiers in Urology 3 (2023): 1296349.

2.

The details of the proposed attention-based network architecture should be given, as it is the basis, and the major forte the proposed pre-void alarm system. References for the proposition of such an architecture are required, as well as the better explanation of modification of the existing CNN LSTM architecture. There are CNN-LSTM networks used for three dimensional data, and the references should be included.

In Figure 6. Filters named “Zhang et.al” and the “proposed filters” are not explained in the text, which is mandatory for anyone interested in the reconstruction or improvement of this 3D data processing network.

Author Response

Thank you for recognizing the value of our study and for recommending an expansion of our literature review. 

We briefly introduced existing works in related fields and added details of them, including what you suggested for us, and comparison between these works and our work to a new section (2. Existing Works in Portable Systems for Bladder Monitoring; p3-4).

Additionally, in response to your valuable comments about our proposed neural network architecture, we elaborate the details of the architecture while citing the works we referenced (p9-10).

Lastly, we clarified what we've referred to Zhang et.al.'s work. We adopted the concept of their work, without implementing it exactly. In addition, we decided to discard the term "proposed filters" in order not to make the readers confused, since the filters (i.e., kernels) used in CNN layer were the general ones, not specifically designed by ourselves. Our intention was that our filters should place separately according to the type of input data because our prediction model has a multi-dimensional structure.

Reviewer 3 Report

Comments and Suggestions for Authors

Attached

Comments on the Quality of English Language

Attached

Author Response

We deeply appreciate your constructive critique and recognize its value in guiding our research forward. We list up what and how we revised our manuscript according to your valuable comments one by one.

Comment 1. The abstract must be strengthened by highlighting the key parameters.

Response: After we revised the manuscript considering all your comments, we summarized the previous version of abstract to emphasize what we have done in this work. You can see the revised version on Page 1.

Comment 2. The introduction section needs to be reframed and elaborated with terms like smart wearable devices and biosensors incorporating ML/AI by adding proper justification and relevant references.

Response: We restructured the Introduction section (p1-3), resulting in adding a new section titled Existing Works in Portable Systems for Bladder Monitoring (p3-4). Our intention was to strengthen our justification and to briefly introduce existing works in related fields in Introduction. Then, we aimed to describe the details of these works, including newly cited ones, and to compare between these works and our work. 

Comment 3. What is the significance of nocturnal enuresis in accordance with wearable systems?

Response: We clearly mentioned how the wearable systems can play a significant role in the context of nocturnal enuresis in the second paragraph of Introduction (p 1-2)

Comment 4. What is the novelty of this proposed work not clearly mentioned? Numerous works are reported in this area of research, how this approach is different. Please justify.

Response: We justify the difference of our work compared to the existing studies in the revised Introduction (p1-3) and Section 2, which was newly added (p3-4). Additionally, we clarify four contributions of this work at the end of Introduction (p3) 

Comment 5. Authors should consider adding these relevant references: 

  • doi.org/10.1145/3341162.3349313
  • doi.org/0.1109/JTEHM.2023.3336889
  • doi.org/10.1016/j.talanta.2024.125817

Response: We cited these works properly across the entire manuscript, including Introduction (p1-3) and Section 2 (p3-4). 

Comment 6. Page 3, lines 87-95 should be converted as a paragraph

Response: We separated these lines as a single paragraph, which is the last paragraph of Introduction (p3)

Comment 7. Figure 2 looks too crammed and needs to be improved.

Response: We renewed Figure 2 (p5) to clearly deliver how Data Collection Supervisor works while discarding redundant or unnecessary details.

Comment 8. Section 2, please include the consumables details with the vendor, specs, and place as well

Response: We added the vendor, specs, and place of the commercial devices (p4 and P5)

Comment 9. Page 5, Algorithm 1 is not required and can be removed

Response: We discarded it since we totally agreed with your suggestion.

Comment 10. Authors should discuss the existing wearable sensors that are useful for human health monitoring with some examples

Response: We added a discussion about how our proposed wearable system should be improved, considering your suggestion, at the paragraph starting with “Thirdly,” on Page 16. 

Comment 11. In Figure 8, the resolution is poor, and the legend font needs to be increased.

Response: We processed the image in a way to satisfy your suggestion (Figure 8; p11).

Comment 12. What were the dynamic range, sensitivity, and reproducibility in the reported literature? Please highlight the key parameters by adding the comparison table.

Comment 13. Add the table showing the comparison study with various parameters

Response: Unfortunately, there was not sufficient information regarding dynamic range, sensitivity, and reproducibility from the existing works that we surveyed. Instead, we added a comparison table (Table 1; p3) briefly showing the key parameters  to be examined for justifying our work. 

Comment 14. Please discuss the challenges and limitations of this work

Response: We carefully improved our Discussion section to address five issues about the challenges and limitations that we discovered and acknowledged. You can see these issues from the paragraph starting with “However,~” to the end of the section (p15-17)

Comment 15. Authors should take care of singular and plural sentences

Comment 17 Please check the English language and improve accordingly

Response: During this revision round, we reviewed overall sentences throughout the entire manuscript to improve English writing.

Comment 16 Please provide the SWOT analysis

Response: We appreciate your valuable suggestion. However, we think that our revised manuscript can provide sufficient contents, including the previous literature review, the description of the proposed framework, the experimental demonstration, and the acknowledged limitations that can be addressed in future works. Through these contents, we believe that we can deliver strengths and weaknesses of our work, which are key results of SWOT analysis. Furthermore, our review of the literature in related fields indicates that conducting a SWOT analysis is not a common practice among papers reporting similar research. We respectfully request your understanding regarding our decision to follow this precedent, as we believe our methodology aligns well with the established practices in our field.

Round 2

Reviewer 2 Report

Comments and Suggestions for Authors

The paper has been revised appropriately and I have no further comments.

Author Response

Thank you for confirming that our revised manuscript has no issues to be updated.

Reviewer 3 Report

Comments and Suggestions for Authors

I went through the improved version of the revised manuscript thoroughly; the authors have significantly enhanced the quality of the technical content; I appreciate their efforts and time. However, still needs some essential factors to be included in the manuscript to look more interesting and exciting, below are my comments:

1) Introduction needs to be comprehensively discussed with cutting-edge technologies such as microfluidics, machine learning, and artificial intelligence in drug delivery applications. The advantages and disadvantages. 

2) Authors claim that they have addressed comment 5 upon checking seems like missing the recommended relevant references: 10.1109/JTEHM.2023.3336889; doi.org/10.1016/j.talanta.2024.125817

3) Authors should address comment 12 separately and Table 1 is not sufficient to justify that the proposed work is novel. Please compare with the recent state-of-the-art (adding 2-3 years articles) in a table.

4) Add more recent references, and remove the older ones, in the proposed area (covering 1-2 years) ~45 references that support your work.

5) Comment 16, I understand the concern, but the author should consider illustrating or portraying the strengths, weaknesses, opportunities, and threats of the proposed work by adding a figure.

Author Response

1) Introduction needs to be comprehensively discussed with cutting-edge technologies such as microfluidics, machine learning, and artificial intelligence in drug delivery applications. The advantages and disadvantages. 

Response 1

Thank you for your valuable comment which can improve our introduction section more informative. We discussed the wide application of AI to various fields in the fourth paragraph of Introduction (p4). We highlighted the added sentences as purple color.

2) Authors claim that they have addressed comment 5 upon checking seems like missing the recommended relevant references: 10.1109/JTEHM.2023.3336889; doi.org/10.1016/j.talanta.2024.125817

Response 2

We’re sorry, and we appreciate your kind notice about our mistakes in adding the relevant references. We properly added the two references in section 2. Existing Works in Portable Systems for Bladder Monitoring (p3-4). We highlighted the section including the two references as green color. 

3) Authors should address comment 12 separately and Table 1 is not sufficient to justify that the proposed work is novel. Please compare with the recent state-of-the-art (adding 2-3 years articles) in a table.

Response 3-1

We clearly inform dynamic range, sensitivity, and reproducibility of our developed NEwear in section 3.1. Wearable devices: NEwear and Commercial Sensor Bands. We highlighted the added description as green color.

Response 3-2

In addition, we agree that there were some outdated references in Table 1, so that we added current references while replacing the outdated ones. We focused on the recent SOTA articles (within 2-3 years), as you requested, but some articles are within 3-4 years as well. We highlighted the added description as green color (p3-4).

4) Add more recent references, and remove the older ones, in the proposed area (covering 1-2 years) ~45 references that support your work.

To satisfy your suggestion, we replaced the outdate references used in the proposed area (i.e., section 3. Materials and Methods) with the recent SOTA articles (within 1-2 years). The references from more previous years should be the fundamental papers in the fields related to disease (i.e. NE) or our previous works. We highlighted the sentences corresponding to the replaced references as blue color (p6-10).

5) Comment 16, I understand the concern, but the author should consider illustrating or portraying the strengths, weaknesses, opportunities, and threats of the proposed work by adding a figure.

We added a figure for SWOT analysis and described it in the first paragraph of Discussion (p16-17).